# Trajectory Tracking of Flexible-Joint Robots Actuated by PMSM via a Novel Smooth Switching Control Strategy

**Yue Wang, Haisheng Yu \*, Jinpeng Yu, Herong Wu and Xudong Liu**

School of Automation, Qingdao University, Qingdao 266000, China; wang1693yue@163.com (Y.W.);
yjp1109@126.com (J.Y.); wu.hr@163.com (H.W.); xudong19871982@163.com (X.L.)
**\*** Correspondence: yu.hs@163.com

**Abstract:** This paper presents the trajectory tracking control of a flexible-joint manipulator driven by permanent magnet synchronous motor (PMSM). Combining the PMSM electrical equation and mechanical equation of robotic manipulators, a novel smooth switching control scheme is proposed. Firstly, the position loop controller of the system is designed, with an improved hierarchical sliding mode control (IHSMC) algorithm proposed to further the response speed of the system, additionally, a robust interconnection and damping assignment passivity-based controller (IDA-PBC) is designed to improve the steady state performance of the system. Then, the IDA-PBC control strategy is leveraged to design the current loop controller of the system, on which basis a hybrid controller with smooth switching is designed. Furthermore, Gaussian function is applied as the smooth switching function of the hybrid controller to promote the switching performance. As a result, the hybrid controller has both good dynamic and steady performance. The simulation results verify the effectiveness of the algorithms.

**Keywords:** flexible-joint robot; PMSM; IHSMC; PCH

## 1. Introduction

Flexible-joint robots are widely used in industrial fields, with a broad application prospect such as aerospace, defense, and medical [1–5]. The articulated flexible robot mostly adopts the harmonic reducer as its transmission mechanism. The harmonic reducer not only has the advantages of small volume, high transmission precision, and high bearing capacity, but also brings joint flexibility into the robots. The introduction of joint flexibility increases the difficulty of control Permanent magnet synchronous motor (PMSM), which has been widely used in the servo system of flexible-joint robots due to their small torque ripple, large torque inertia ratio, and high efficiency. In the past, the dynamic model of servo motors as actuators is often neglected in the controller design of mechanical systems [6,7]. The reason for this is that considering the actuator complicates the design of the controller. However, regardless of the dynamics model of the actuator, the performance of the closed-loop system may be degraded or even have instability [8].

Most scholars have studied how to better combine electrical dynamics with mechanical dynamics, on which basis the position tracking controller of the robot is designed. An adaptive fuzzy voltage control strategy for a DC-driven rigid-joint robot with uncertainty compensation is proposed in [9]. In [10], an observer-based robust control strategy for DC-driven flexible joint robots without speed sensors is proposed. An adaptive fuzzy control strategy for a PMSM-driven rigid-joint robot is proposed in [11], but this article did not make a specific analysis of the controller performance. A global proportion–integral–derivate (PID) control strategy for a PMSM-driven rigid-joint robot

and a global stability proof method of PID control are proposed in [12]. Obviously, PMSM-driven flexible-joint robots have been rarely researched until now.

This paper presents a hybrid controller for PMSM-driven flexible-joint robots. In order to make the control system take into account both good dynamic and steady state performance, some researchers have begun to propose to combine two control strategies with different points through smooth switching, to achieve a hybrid control strategy based on time switching [13,14]. However, it is difficult to determine the switching time point, which is one of the disadvantages of such control strategies. In addition, it is difficult to ensure the control effect when facing unknown disturbances by using the hybrid control strategy with time function as the switching function. Therefore, in this paper, a Gaussian switching function based on velocity error is proposed to realize smooth switching of hybrid controllers. The hybrid controller mainly includes a sliding mode controller and port-controlled Hamiltonian (PCH) controller. Sliding mode control is widely used in flexible manipulators due to its excellent robustness and excellent dynamic performance [15,16]. Therefore, an improved hierarchical sliding mode control strategy is designed to improve the dynamic performance of the system. The chattering cannot be eliminated due to sliding mode control. Hence, IDA-PBC control is designed for the system, with good steady-state performance. As we all know, IDA-PBC has a good control effect on systems whose dynamics are difficult to accurately model [17–19]. The state error PCH system of the flexible joint is constructed to achieve better tracking effect [20]. The added PID outer ring control can effectively reduce the influence of external disturbance on the system and improve the steady state performance of the system [21,22]. On this basis, the electrical subsystem of the PMSM is written in the form of PCH, and the IDA-PBC control is designed for current loop control.

This paper is organized as follows. In Section 2, the system dynamics model proposed in this paper is established. In Section 3, the IHSMC controller and PID-IDA-PBC are designed for the mechanical subsystem, and the IDA-PBC controller is designed for the electrical subsystem. Based on this, a hybrid controller is proposed. Simulation results and analysis are presented in Section 4.

## 2. Dynamic Model of Robots

The dynamic model of an n-degree-of-freedom flexible-joint robot driven by a permanent magnet synchronous motor can be expressed as

$$\begin{cases} M(q)\ddot{q} + C(q,\dot{q})\dot{q} + G(q) = K(r\theta - q) + D(r\dot{\theta} - \dot{q}) \\ J_m\ddot{\theta} + K(r\theta - q) + D(r\dot{\theta} - \dot{q}) = r\tau_m \\ L_d\frac{di_d}{dt} + R_s i_d - n_p\dot{\theta}L_q i_q = u_d \\ L_q\frac{di_q}{dt} + R_s i_q + n_p\omega L_d i_d + n_p\omega\Phi = u_q \end{cases} \tag{1}$$

$$\tau_m = n_p(L_d - L_q)i_d i_q + \Phi i_q \tag{2}$$

where the mechanical subsystem dynamics model is derived from the model proposed in [23], and the electrical subsystem model is the electrical equation of the PMSM in the d–q coordinate system. $q$ is the vector of joint positions, $\theta$ is the motor position vector, $M(q) \in R^{n \times n}$ is the inertial matrix, $C(q,\dot{q}) \in R^{n \times n}$ is the Coriolis force and centripetal force matrix, $G(q) \in R^n$ is the gravity torque vector, $K = diag\{K_i\} \in R^{n \times n}$ is the robotic joint stiffness matrix, $D = diag\{D_i\} \in R^{n \times n}$ is the robotic joint damping matrix, $J_m = r^2 J'_m = diag\{J_{mi}\} \in R^{n \times n}$ is the constant diagonal matrix, $J'_m$ is the motor inertia matrix, $L_d, L_q, R_s, \Phi$, and $n_p$ are constant diagonal matrices, which represent the inductance, stator resistance, flux linkage, and pole pair of PMSM, $u_q = [u_{q1}, u_{q2}, ...,]^T \in R^n$, $u_d = [u_{d1}, u_{d2}, ...,]^T \in R^n$ are the stator voltage at the q and d axes, respectively, $i_q = [i_{q1}, i_{q2}, ...,]^T \in R^n$, $i_d = [i_{d1}, i_{d2}, ...,]^T \in R^n$, are the stator voltage at the q and d axes, correspondingly, and $r$ is the reduction ratio.

According to [19], the basic properties of some mechanical subsystems can be obtained to facilitate control the design.

**Property 1.** *The matrix is symmetric and positive definite.*

**Property 2.** $\dot{q}^T[\dot{M}(q) - 2C(q, \dot{q})]\dot{q} = 0, \forall q, \dot{q} \in R.$

## 3. Hybrid Control Design and Analysis

In this section, we uses double closed-loop control. The outer loop is a position loop, and two position controllers are designed according to the IHSMC and PID-IDA-PBC algorithms, respectively. In order to realize the advantages of the two control algorithms, the Gaussian function is adopted as the switching function, and the smooth switching is performed according to the rate of change of the position error. For the purpose of simplifying the controller design, we use a surface PMSM as the servo motor. In order to achieve current tracking control, this paper will use the IDA-PBC control strategy as the current controller. The system diagram of the hybrid control system is as follows in Figure 1.

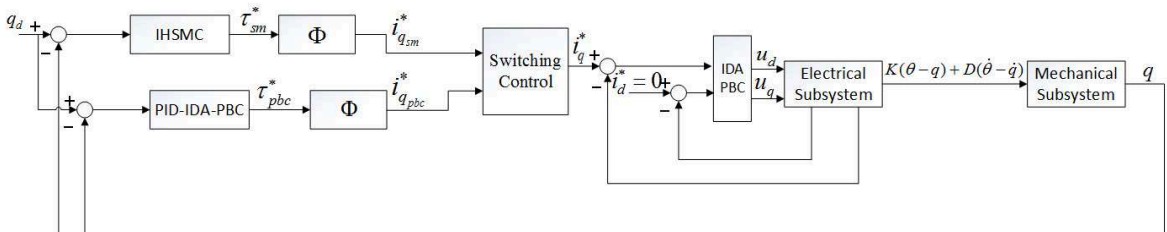

**Figure 1.** The figure of the hybrid control system.

### 3.1. Hierarchical Sliding Mode

The paper [2–5,24] proposes an improved hierarchical sliding mode control. Firstly, the first sliding surface is improved, and a nonlinear super-curved sliding surface is adopted. This sliding surface is superior to the traditional linear sliding surface, so that the system can reach the equilibrium point at a higher speed. Then, the secondary sliding surface is designed on the basis of the first sliding surface. Finally, a double power reaching law is designed and the asymptotic stability of the subsystem is proved [25].

Define $q_d$ as the desired angular position of the link, $\theta_d$ is the desired angular position of the motor. Then define the state error as $q_e = q - q_d$, $\theta_e = \theta - \theta_d$. Designing the first layer sliding surface $s_q, s_\theta$ as

$$s_q = \dot{q}_e + a_1 q_e{}^{q_1/p_1} + b_1 q_e{}^{p_2/q_2} \tag{3}$$

$$s_\theta = \dot{\theta}_e + a_2 \theta_e{}^{q_3/p_3} + b_2 \theta_e{}^{p_4/q_4} \tag{4}$$

where $a_1 > 0, a_2 > 0, b_1 > 0, b_2 > 0$ are constant, $p_1 > q_1 > 0, p_2 > q_2 > 0, p_3 > q_3 > 0, p_4 > q_4 > 0$ are odd.

The following simulation results verify that the nonlinear hyperboloid sliding surface application has a good application effect in the layered sliding mode controller. On the basis of a first layer of sliding surface, the second layer sliding surface is designed.

$$s = \alpha s_q + \beta s_\theta \tag{5}$$

where $\alpha, \beta$ are the constants to be designed.

The double power reaching law is defined as follows

$$\dot{s} = -r_1 \|s\|^{k_1} \text{sgn}(s) - r_2 \|s\|^{k_2} \text{sgn}(s) - \frac{k_3}{\|s\| + k_4}(s_q^T \dot{s}_q) \tag{6}$$

where $\|s\| > 0$, $k_1 > 1, 1 > k_2 > 0$, $r_1, r_2$ are the constants to be designed, depending on the requirements of the system's disturbance–resistance performance, $k_3$ is a positive integer, $k_4$ is a positive number that approaches to zero.

From Equations (3)–(5)

$$\dot{s}_q = \ddot{q}_e + a_1 \frac{q_1}{p_1} \dot{q}_e^{q_1/p_1} + b_1 \frac{p_2}{q_2} \dot{q}_e^{p_2/q_2} \tag{7}$$

$$\dot{s}_\theta = \ddot{\theta}_e + a_2 \frac{q_3}{p_3} \dot{\theta}_e^{q_3/p_3} + b_2 \frac{p_4}{q_4} \dot{\theta}_e^{p_4/q_4} \tag{8}$$

$$\dot{s} = \alpha \dot{s}_q + \beta \dot{s}_\theta \tag{9}$$

Substituting Equations (6)–(8) into Equation (9) yields

$$\begin{aligned}
\tau_{sm} = &-\frac{\alpha J_m}{\beta} \left\{ M^{-1}(q) \left[ K(\theta - q) + D(\dot{\theta} - \dot{q}) - C(q, \dot{q})q - G(q) \right] - \ddot{q}_d + a_1 \frac{q_1}{p_1} \dot{q}_e^{q_1/p_1} + b_1 \frac{p_2}{q_2} \dot{q}_e^{p_2/q_2} \right\} \\
&+ \frac{J_m}{\beta} \left[ -r_1 \|s\|^{k_1} \text{sgn}(s) - r_2 \|s\|^{k_2} \text{sgn}(s) - \frac{k_3}{\|s\|+k_4} (s_q^T \dot{s}_q) \right] \\
&+ K(\theta - q) + D(\dot{\theta} - \dot{q}) + J_m(\ddot{\theta}_d - a_2 \frac{q_3}{p_3} \dot{\theta}_e^{q_3/p_3} - b_2 \frac{p_4}{q_4} \dot{\theta}_e^{p_4/q_4})
\end{aligned} \tag{10}$$

The Lyapunov function is defined as

$$\begin{aligned}
V_{HSM} &= \frac{1}{2} s^T s + \frac{1}{2} s_q^T s_q \\
\dot{V}_{HSM} &= s^T \dot{s} + s_q^T \dot{s}_q \\
&= s^T \dot{s} + s_q^T \dot{s}_q \\
&= s^T (\alpha \dot{s}_q + \beta \dot{s}_\theta) + s_q^T \dot{s}_q \\
&= -s^T \left( r_1 \|s\|^{k_1} \text{sgn}(s) + r_2 \|s\|^{k_2} \text{sgn}(s) + \frac{k_3}{\|s\| + k_4} (s_q^T \dot{s}_q) \right) + s_q^T \dot{s}_q
\end{aligned} \tag{11}$$

According to [24], the following properties can be obtained

**Property 3.** *Provided on the outer bound of unknown disturbance is $M_D$, when $k_3 > 1$, the second layer sliding surface is bounded stable. s can converge to any small area $\Omega$ for a limited time. $\Omega$ can be expressed as*

$$\Omega = \left\{ s \middle| \|s\| \le \max \left( \left( \frac{M_D}{r_1} \right), \left( \frac{M_D}{r_2} \right), k_4 \right) \right\} \tag{12}$$

**Property 4.** *If s is bounded then $s_q$ is bounded.*

Therefore, the first layer sliding surface and the second layer sliding surface are both bounded and stable.

*3.2. PID-IDA-PBC*

In this section, the mechanical subsystem table is first written in the form of a PCH. Then, the state error PCH system is constructed by the IDA-PBC control strategy, and the PID outer loop controller is designed to improve the anti-interference ability and steady state performance of the control system. Finally, the asymptotic stability of the subsystem is proved. The dynamics model of the mechanical subsystem can be expressed in the form of a PCH system as follows [19]

$$\begin{cases} \dot{x} = [\mathcal{J}(x) - \mathcal{R}(x)] \frac{\partial H(x)}{\partial x} + g(x) \tau_{pbc} \\ y = g^T(x) \frac{\partial H(x)}{\partial x} \end{cases} \tag{13}$$

where $x = (q, \theta, p_q, p_\theta)^T$. Hamiltonian function is set as

$$H(q, \theta, \, p_q, p_\theta) = \frac{1}{2} p_q^T M^{-1}(q) p_q + \frac{1}{2} p_\theta^T J_m p_\theta + V_g(q) + V_k(q, \theta) \tag{14}$$

where $p_q = M(q)\dot{q}$, $p_\theta = J_m\dot{\theta}$ are momentum, $V_g(q)$ is gravitational potential energy, $V_k(q, \theta) = \frac{1}{2}K(\theta - q)^2 + \frac{1}{2}D(\dot{\theta} - \dot{q})^2$ is elastic potential energy, $J_m$ is a positive number that is not zero.

Define the state error of the system as $\tilde{x} = x - x_d$. Then, the feedback control law $\tau'_{pbc} = \beta(x)$ is obtained by the IDA-PBC control strategy, and the closed-loop state error PCH system is obtained as follows [20]

$$\dot{x} = [\mathcal{J}_d(\tilde{x}) - \mathcal{R}_d(\tilde{x})]\frac{\partial H_d(\tilde{x})}{\partial \tilde{x}} \tag{15}$$

where $J_d(\tilde{x})$ is a skew-symmetric matrix, $R_d(\tilde{x})$ is a positive semi-definite matrix.

The new Hamiltonian function is set as $H_d(\tilde{x}) = \frac{1}{2}\tilde{p}_q^T M_d^{-1}(\tilde{q})\tilde{p}_q + \frac{1}{2}\tilde{p}_\theta^T J_{md}^{-1}\tilde{p}_\theta + V_g(q) + V_k(q, \theta) - V_{\tilde{g}}(\theta)$. System (15) is a PCH system that is asymptotically stable at equilibrium point $\tilde{x} = 0$. The designed state feedback control law can be expressed as

$$\begin{aligned} \tau'_{pbc} = {}& K(\theta - q) + D(\dot{\theta} - \dot{q}) - \lambda_1 M_d^{-1}(\tilde{q})\tilde{p}_q - \lambda_2 J_{md}^{-1}\tilde{p}_\theta \\ & - \frac{J_{md}}{J_m}\left[K(\theta - q) + D(\dot{\theta} - \dot{q}) + k_d\tilde{q} - g(q)\right] \end{aligned} \tag{16}$$

where $\lambda_1, \lambda_2, J_{md}, k_d$ are positive diagonal matrix, $J_{md}$ is a positive number that is not zero.

In this paper, in order to improve the steady state performance and the disturbance–resistance performance of the system. We consider the system (15) to be expressed as [22]

$$\dot{x} = [\mathcal{J}(\tilde{x}) - \mathcal{R}(\tilde{x})]\frac{\partial H_d(\tilde{x})}{\partial \tilde{x}} + g(x)v \tag{17}$$

where $v$ is is a dynamic controller to improve the anti-interference performance of the system.

Introduce the globally defined transform of coordinates $z = \psi(\tilde{x}, \gamma)$ given by

$$\begin{aligned} z_1 &= \tilde{q} \\ z_2 &= \tilde{\theta} \\ z_3 &= \tilde{p}_q \\ z_4 &= \vartheta(\tilde{x}, \gamma) \\ z_5 &= \gamma \end{aligned} \tag{18}$$

where $\gamma \in R^n$ is the state of the outer loop controller. An extended PCH system consisting of a new variable $z$ can be expressed as

$$\begin{bmatrix} \dot{z}_1 \\ \dot{z}_2 \\ \dot{z}_3 \\ \dot{z}_4 \\ \dot{z}_5 \end{bmatrix} = \begin{bmatrix} & & & 0 \\ & \mathcal{J}_d(\tilde{x}) - \mathcal{R}_d(\tilde{x}) & & -\eta_2 \\ & & & 0 \\ & & & -\eta_1 \\ 0 & \eta_2 & 0 & \eta_1 & -\xi \end{bmatrix} \frac{\partial H_z(z)}{\partial z} \tag{19}$$

$$H_z(z) = \frac{1}{2}z_3^T M_d^{-1}(q)z_3 + \frac{1}{2}z_4^T J_{md}^{-1}z_4 + \frac{1}{2}z_5^T \eta_3 z_5 + V_d(z) \tag{20}$$

where $\xi = J_{md}^{-1} J_m\eta_1\eta_2, \eta_1, \eta_2, \eta_3$ are positive diagonal matrix.

From Equations (15), (17)–(19)

$$z_4 = \vartheta(\tilde{x}, \gamma) = \tilde{p}_\theta + J_m\eta_2\eta_3\gamma \tag{21}$$

According to the Equations (15), (17), (19), and (21), the outer loop control law can be obtained as follows

$$\begin{aligned} v = {}& \lambda_2\frac{\partial H_d}{\partial \tilde{p}_\theta} - \lambda_2\left(J_m\dot{\tilde{\theta}} + \eta_2\eta_3\int\left(\eta_2\frac{\partial H_d}{\partial \tilde{p}_\theta} + \eta_1 J_d^{-1}J\dot{\tilde{\theta}}\right)dt\right) - \eta_1\eta_3\int\left(\eta_2\frac{\partial H_d}{\partial \tilde{p}_\theta} + \eta_1 J_d^{-1}J\dot{\tilde{\theta}}\right)dt \\ & - J_m\eta_2\eta_3\left(\eta_2\frac{\partial H_d}{\partial \tilde{p}_\theta} + \eta_1 J_d^{-1}J\dot{\tilde{\theta}}\right) \end{aligned} \tag{22}$$

According to the Equations (16), (17), and (22), the PID-IDA-PBC law can be expressed as

$$
\begin{aligned}
\tau_{pbc} = {}& \frac{\partial H}{\partial \tilde{\theta}} - \frac{J_{md}}{J_m}\frac{\partial H_d}{\partial \tilde{\theta}} - \lambda_1 \frac{\partial H_d}{\partial \tilde{p}_d} - \lambda_2 \left( J_m \dot{\tilde{\theta}} + \eta_2\eta_3 \int \left( \eta_2 \frac{\partial H_d}{\partial \tilde{p}_\theta} + \eta_1 J_d^{-1} J\dot{\tilde{\theta}} \right) dt \right) \\
& - \eta_1\eta_3 \int \left( \eta_2 \frac{\partial H_d}{\partial \tilde{p}_\theta} + \eta_1 J_d^{-1} J\dot{\tilde{\theta}} \right) dt - J_m\eta_2\eta_3 \left( \eta_2 \frac{\partial H_d}{\partial \tilde{p}_\theta} + \eta_1 J_d^{-1} J\dot{\tilde{\theta}} \right)
\end{aligned}
\tag{23}
$$

The Lyapunov function of the subsystem is defined as Hamiltonian function

$$
V_{PCH} = H_z(z) > 0
$$

$$
\dot{V}_{PCH} = \dot{H}_z(z) = \frac{dH_z(z)}{dt} = -\left[ \frac{\partial H_z(z)}{\partial z} \right]^T R(z) \frac{\partial H_z(z)}{\partial z} \leq 0
$$

According to LaSalle's invariance principle, if the largest invariant set of the closed-loop system is included in the set

$$
\left\{ \dot{z} \in R^n \,\middle|\, \left[ \frac{\partial H_z(z)}{\partial z} \right]^T R(z) \frac{\partial H_z(z)}{\partial z} = 0 \right\}
$$

then subsystem is asymptotically stable at the point of $z = 0$.

### 3.3. Design of Current Controller

The research found that the design and parameter selection of the current loop controller also affects the control accuracy of the system. According to the literature [26,27], in order to obtain a better tracking effect of the current loop, the electrical subsystem of the system (1) is expressed as the PCH form

$$
\begin{cases}
\dot{x} = [\mathcal{J}_e(x) - \mathcal{R}_e(x)]\frac{\partial H_e(x)}{\partial x} + g(x)u \\
y = g^T(x)\frac{\partial H_e(x)}{\partial x}
\end{cases}
\tag{24}
$$

$$
H_e = \frac{1}{2}(Li_d^2 + \frac{1}{2}Li_q^2)
$$

The IDA-PBC and energy shaping control technique are used to find the feedback control to form a closed-loop system as [19]

$$
\dot{x} = [J_{ed}(x) - R_{ed}(x)] \frac{\partial H_{ed}(x)}{\partial x}
\tag{25}
$$

$$
H_{ed} = \frac{1}{2}(L\tilde{i}_d^2 + \frac{1}{2}L\tilde{i}_q^2)
$$

where $J_{ed}(x)$ is desired interconnection matrix, $R_{ed}(x)$ is desired damping matrix, and

$$
\begin{aligned}
J_{ed} &= -J_{ed}^T = J_e + J_{ea} \\
R_{ed} &= R_{ed}^T = R_e + R_{ea} \geq 0
\end{aligned}
\tag{26}
$$

$$
J_{ea}(x) = \begin{bmatrix} 0 & \mu \\ -\mu & 0 \end{bmatrix}, R_{ea}(x) = \begin{bmatrix} \mu_1 & 0 \\ 0 & \mu_2 \end{bmatrix}
$$

where $\mu$ is the desired interconnection parameter; $\mu_1$, $\mu_2$ are the desired damping parameter.

From Equations (24)–(26) the Hamiltonian current controller can be derived as

$$
\begin{aligned}
u_d &= R_s i_d^* + n_p \omega L_q i_q^* + \mu_1(i_d^* - i_d) - \mu(i_q^* - i_q) \\
u_q &= R_s i_q^* + n_p \omega L_d i_d^* + n_p \omega \Phi + \mu_2(i_q^* - i_q) + \mu(i_d^* - i_d)
\end{aligned}
\tag{27}
$$

According to [26,27], the electrical subsystem is globally asymptotically stable.

*3.4. Hybrid Control Strategy*

As indicated above, both different position control methods have advantages and characteristics. IHSMC controller has fast response speed and good dynamic performance, but the problem is that it is difficult to eliminate chattering. The PID-IDA-PBC controller has high accuracy and strong disturbance effects, but it is difficult to take into account both dynamic and steady performance. In this section, we propose a smooth switching strategy based on velocity error to complement the advantages of the two control strategies. In order to avoid the instability of the system in the switching point neighborhood, a Gaussian function is introduced as a smooth switching factor. Compared with the problem that the switching point is difficult to be determined in previous studies [13,14,28], the rate of change of velocity can be used as the switching index to solve this problem. Then the smooth switching function

$$f(\alpha) = e^{-\frac{(\alpha-l)^2}{2c^2}} \tag{28}$$

where $\alpha = |\dot{e}| = |\dot{q} - \dot{q}_d|$ is rate of position error, $c$ is a constant, $l$ is upper limit of speed error switching.

Then hybrid control strategy can be expressed as

$$\begin{cases} i_{qi}^* = i_{qsm_i}^* & h_i \le \alpha_i \\ i_{qi}^* = (1 - f(\alpha_i)) i_{qsm_i}^* + f(\alpha_i) i_{qpbc_i}^* & l_i \le \alpha_i \le h_i \\ i_{qi}^* = i_{qpbc_i}^* & \alpha_i \le l_i \end{cases} \tag{29}$$

where $h$ is upper limit of speed error switching, $i = 1, \ldots, n$.

The use of the position error change ratio as a basis for smooth switching is considered to be that the faster the error convergence speed for a stable system, the higher its response rate. For the trajectory tracking system, a higher the position error change ratio means a faster the response rate and when the system enters the steady-state, we expect the position error to change slowly and gradually approach zero. Hence, the switching interval is selected and the size of the switching function parameter is adjusted according to the rate of change of the position error. In the steady state, due to the high-frequency small chattering caused by the disturbance, the hybrid control process is newly entered until the chattering is eliminated. Hence, when $h_i \ge \alpha_i$, the IHSMC control strategy is adopted to improve the dynamic performance of the system; when $\alpha_i \le l_i$, the PID-IDA-PBC control strategy is used to improve the control accuracy and anti-interference ability of the system; when $l_i < \alpha_i < h_i$, the smoothing switch is realized by smoothing function.

## 4. Simulation and Comparative Analysis

In this section, we will verify the control algorithm proposed in Section 3 by simulation. In order to facilitate the simulation, we consider 2DOF flexible manipulator as the controlled object. Firstly, the IHSMC algorithm proposed in this paper is simulated and compared with the traditional HSMC algorithm. The superiority of the control strategy is verified. Then, a PID-IDA-PBC proposed in this paper is simulated and implemented with the IDA-PBC algorithm. In comparison, it is verified that the PID-IDA-PBC algorithm has better steady state performance and disturbance–resistance ability. In the end, we proceed simulations using hybrid control. We use the parameter values of robot manipulator reported in [29]. In this simulation we assume that each joint of the flexible-joint robot is driven by PMSM. The parameters of PMSM are given as $R_s = 0.901\ \Omega$, $L_q = L_d = 0.0065$ H, $n_p = 4$, $\Phi = 0.031$ Wb, $J = 0.00012$ Kg·m², $r = 0.01$. The initial state is set as $q_1(0) = q_2(0) = \theta_1(0) = \theta_2(0) = 0$, we require the expected position vector to be $q_d = [\sin t, \sin t]^T$, $\theta_d = K^{-1}g(q) + q_d$. Lastly, we agree that the counterclockwise direction is the positive direction of the rotation of robot.

For the IHSMC algorithm, the selected controller parameters are $a_1 = diag\{1,1\}$, $a_2 = diag\{20,20\}$, $a_3 = diag\{15,14\}$, $b_1 = diag\{0.5,0.1\}$, $b_2 = diag\{0.5,0.1\}$, $b_3 = diag\{30,30\}$, $r_1 = diag\{20,10\}$, $r_2 = diag\{1,0.8\}$, $k_1 = 2$, $k_2 = 0.8$, $k_3 = 2$, $k_4 = diag\{0.00005,0.00005\}$, $p_1 = p_2 = $

9, $p_2 = p_4 = 5$, $q_1 = q_3 = 7$, $q_2 = q_4 = 3$. The parameter selection of the joint 1 current controller are $\mu = 2$, $\mu_1 = \mu_2 = 40$. The original HSMC control law can be expressed as [24]

$$\begin{aligned}
\tau'_{sm} = & -\frac{\alpha J_m}{\beta} \left\{ M^{-1}(q) \left[ K(\theta - q) + D(\dot{\theta} - \dot{q}) - C(q,\dot{q})q - G(q) \right] - \ddot{q}_d + a_1 \dot{q}_e \right\} \\
& + \frac{J_m}{\beta} \left[ -r_1 \|s\|^{k_1} \mathrm{sgn}(s) - r_2 \|s\|^{k_2} \mathrm{sgn}(s) - \frac{2}{\|s\|+k_4}(s_q^T \dot{s}_q) - k_5 s \right] \\
& + K(\theta - q) + D(\dot{\theta} - \dot{q}) + J_m(\ddot{\theta}_d - a_2 \dot{\theta}_e)
\end{aligned} \tag{30}$$

where $k_5 = diag\{10, 10\}$. According to the results shown in Figure 2, joint 1 and joint 2 enter a steady state after a dynamic process of about 0.4 s, but there is still a small chattering in the system at steady state. Compared with the original HSMC algorithm, the IHSMC algorithm has a faster response speed and chattering is effectively suppressed. Figure 3 shows the simulation curves of $u_d$, $u_q$ under IHSMC and HSMC method.

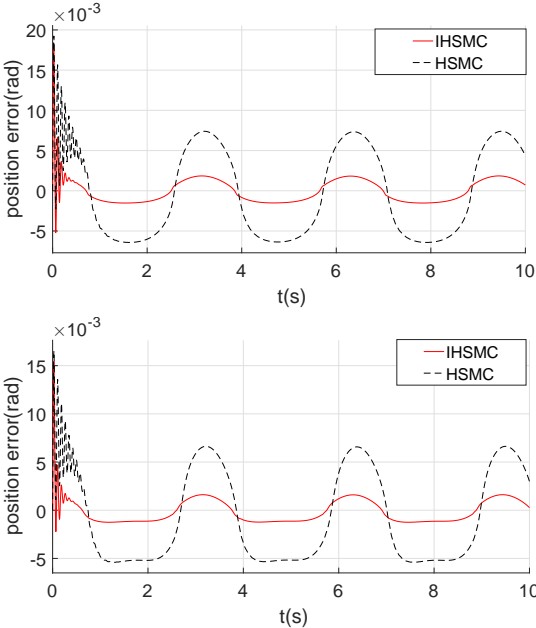

**Figure 2.** Improved hierarchical sliding mode control (IHSMC) and HSMC position tracking comparison curve.

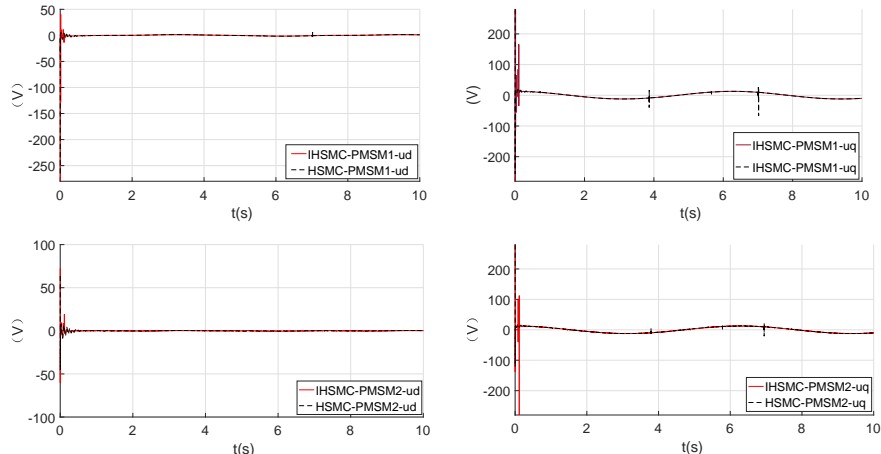

**Figure 3.** Control voltage of each joint motor.

The simulation experiment of the PID-IDA-PBC algorithm is divided into two parts. Firstly, the performance difference between the algorithm and the traditional IDA-PBC algorithm is

compared without disturbance. Then, a constant value disturbance signal of 2 N·m is added at the System dynamic Equation (2) at 4 s to verify the disturbance–resistance performance of the algorithm. Parameters of PID-PBC-PCH controller: $k_d = diag\{1200, 100\}$, $\lambda_1 = diag\{1, 1\}$, $\lambda_2 = diag\{100, 100\}$, $\eta_1 = diag\{0.95, 0.93\}$, $\eta_2 = diag\{1, 1\}$, $\eta = diag\{1.08, 1.02\}$. According to Figure 4, it can be concluded that the PID-IDA-PBC has improved dynamic performance and steady-state performance compared to the conventional IDA-PBC control. The PID-IDA-PBC control algorithm can improve the steady-state performance by adjusting $v$. Figure 5 shows the simulation curves of $u_d, u_q$ under IDA-PBC and PID-IDA-PBC method. At the same time, it will reduce the dynamic performance. In order to validate the robustness of the PID-IDA-PBC algorithm, constant-size perturbations of different sizes are added to the drive motors of joint 1 and joint 2 at 4 s. The result is shown in Figure 6. The PID-IDA-PBC algorithm has excellent disturbance–resistance performance.

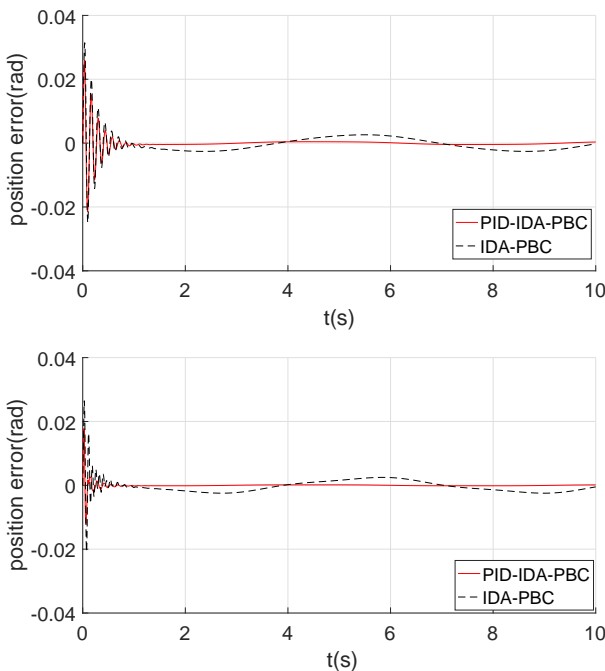

**Figure 4.** Interconnection and damping assignment passivity-based controller (IDA-PBC) and proportion–integral–derivate (PID)-IDA-PBC position tracking comparison curve.

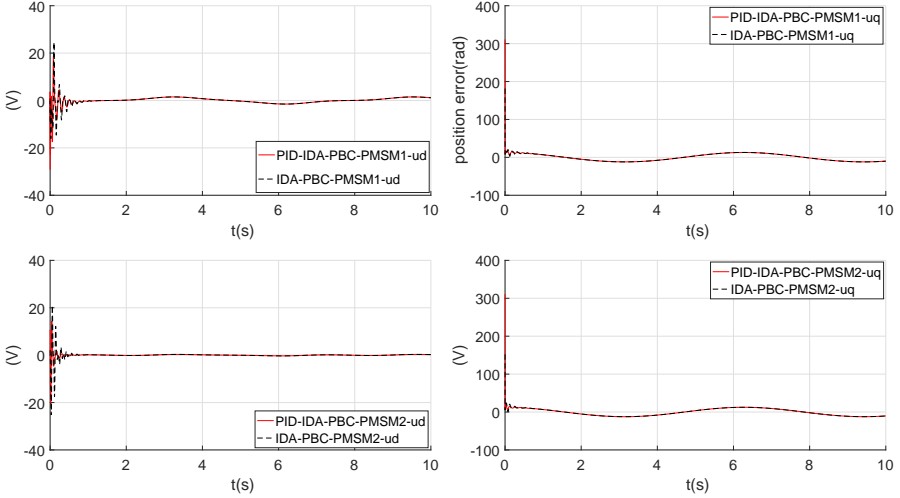

**Figure 5.** Control voltage of each joint motor.

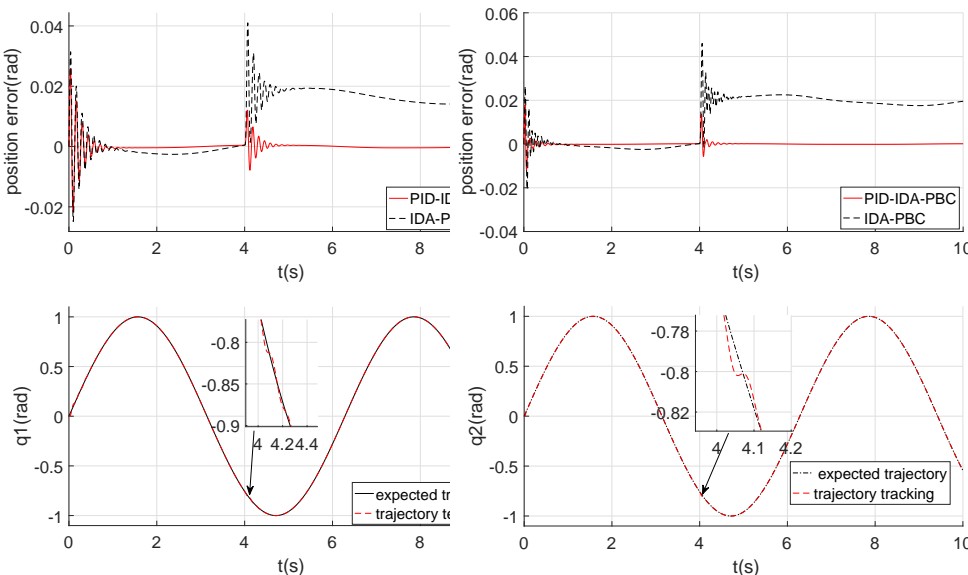

**Figure 6.** Curves with effect of interference of each joint.

According to the above, it can be concluded that a single control algorithm finds it difficult to balance both dynamic performance and steady state performance. Therefore, this paper proposes a hybrid control strategy that attempts to combine the advantages of the two control algorithms. The simulation experiment of the hybrid control strategy was divided into three parts. First, we compared the control effects of hybrid control and single control algorithms. Then, a disturbance was added at 4 s to verify the immunity of the hybrid control. Finally, a simulation experiment was designed to analyze the effect of switching parameters on the effects of hybrid control. The hybrid controller parameter selection is $c_1 = 0.08, c_2 = 0.008, l_1 = 0.03, l_2 = 0.003, h_1 = 0.7, h_2 = 0.2$.

According to the simulation results in Figure 7, it can be concluded that the hybrid control strategy combines the advantages of the two control algorithms well, and has good dynamic performance and high tracking accuracy. Figure 8 shows the simulation curves of $u_d, u_q$ under Hybrid control method. The smoothing function curve in Figure 9 reflects the proportion of PID-IDA-PBC control during the switching process. Figure 10 is the trajectory curves when there is a disturbance imposed at the time of $t = 4$ s. The result shows that even if the system generates small high-frequency chattering due to disturbance, it can eventually eliminate chattering. The simulation results in Figure 11 show that the control effect of the hybrid control strategy is related to the switching parameter $c$. The curve with smooth switching function under different parameters is shown in Figure 12. According to the results in Figures 11 and 12, it can be concluded that the effect of the hybrid control is affected by the parameter $c$ and the switching interval $[l, h]$. The larger the $c$ value, the earlier the position error change ratio is affected by the PID-IDA-PBC algorithm. According to the value of $c$, the appropriate switching interval is selected to achieve smooth switching, and the abrupt change of the switching is avoided.

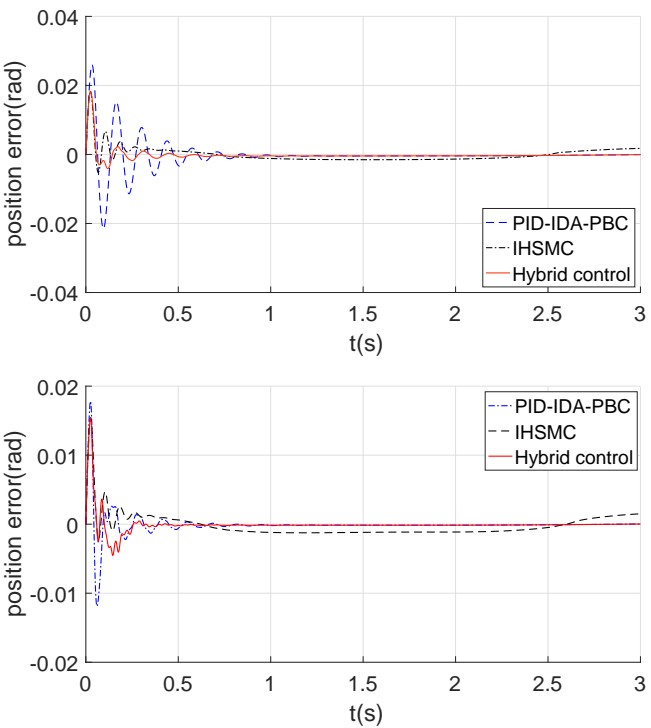

**Figure 7.** Curves with different control algorithm of each joint.

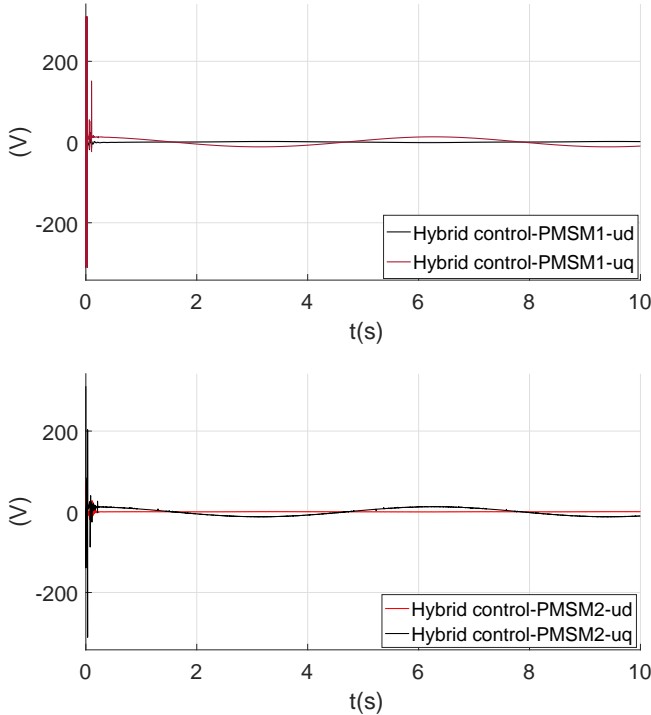

**Figure 8.** Curve with control voltage of each joint motor.

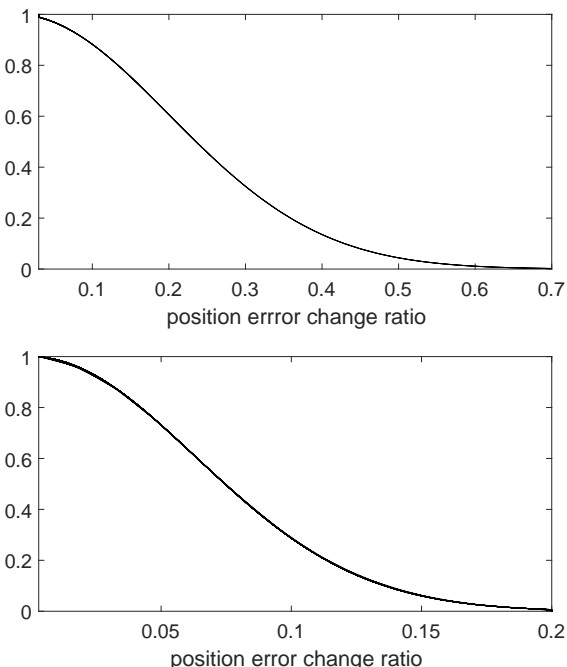

**Figure 9.** Curve with smoothing function.

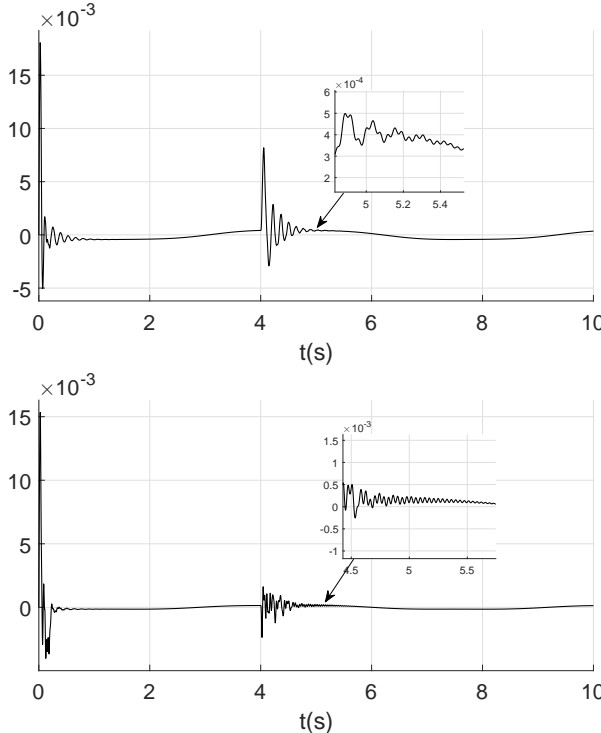

**Figure 10.** Curve with hybrid control tracking error under the influence of disturbance.

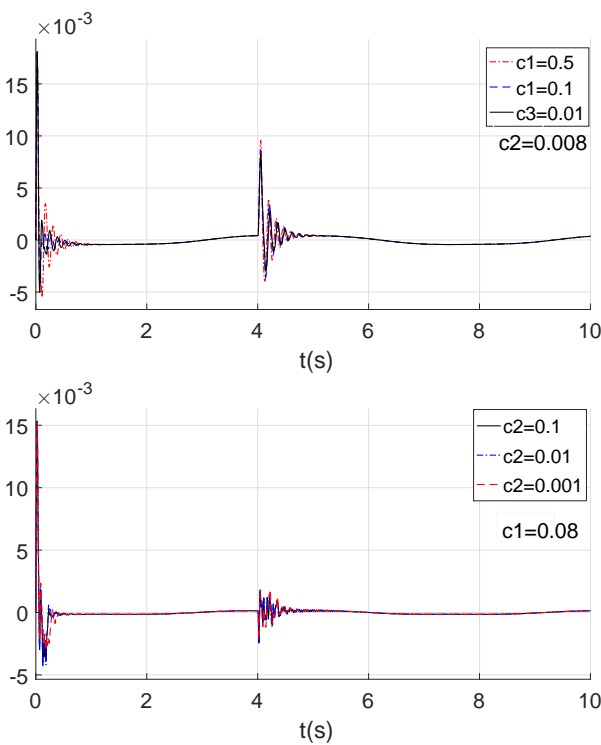

**Figure 11.** Curve with tracking error of hybrid control under the different parameters.

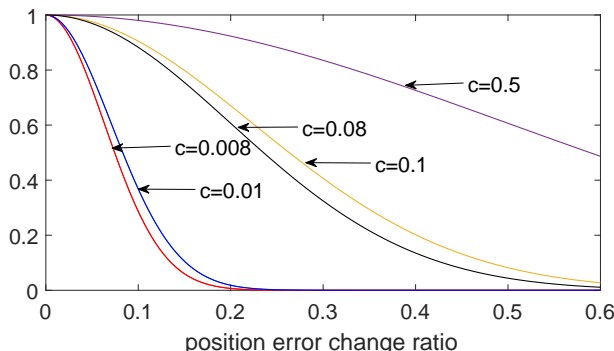

**Figure 12.** Curve with smoothing function under the different parameters.

## 5. Conclusions

This paper proposes a hybrid control strategy based on velocity error for the position tracking control of flexible-joint robots driven by PMSM. The IHSMC algorithm is used to improve the dynamic performance of the system. On the other hand, the PID-IDA-PBC is used to improve the steady-state performance of the system in order to boost its anti-interference ability. In consideration of the influence of the servo driver on the control effect, the IDA-PBC control algorithm is used to control the electrical subsystem controls. Gaussian function is used as the smooth switching function of the hybrid control algorithm. Simulation results illustrate that hybrid control strategy can fully combine the advantages of the two control algorithms, which in turn result in a system with both good dynamic performance and control precision. In brief, the control algorithm has threefold advantages, which are simple structure and application value. In the future research, intelligent algorithms will be used to optimize the switching process to find the optimal switching time (internal) and switching parameters. On the other hand, we will consider how to suppress the sliding mode chattering and reduce the impact of chattering on the system.

**Author Contributions:** Y.W. designed the control strategy, completed the simulation experiment, analyzed the simulation data, and wrote the manuscript. H.Y. and X.L. conceived the structure of the paper, proposed amendments and provided funding support. J.Y. and H.W. proposed amendments and analyzed the simulation data.

**Funding:** This research was funded by the National Natural Science Foundation of China under Grant 61573203.

**Conflicts of Interest:** The authors declare no conflict of interest.

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
