# Peer review of "Trajectory Tracking of Flexible-Joint Robots Actuated by PMSM via a Novel Smooth Switching Control Strategy"

_applsci, doi:10.3390/app9204382_

Round 1

Reviewer 1 Report

This paper proposes a hybrid control strategy based on velocity error for the position tracking control of flexible joint robot driven by PMSM.

In general, authors present hybrid control based on the velocity error and sliding mode. Authors should consider the following comments to clarify the main contributions of their paper.    

1.- In the page 1, in the introduction, authors say "Nowadays, flexible joint robots are widely used in industrial fields, with a broad application prospect in aerospace, defense, medical and other fields[1].", in this part, they should include references [a], [b], [c], [d] which consider flexible joint robots.

[a] Robust feedback linearization for nonlinear processes control, ISA Transactions, Vol. 74, pp. 155-164, 2018.

[b] Modeling and regulation of two mechanical systems, IET Science, Measurement & Technology, Vol. 12, No. 5, pp. 657-665, 2018.

[c] Modified repetitive learning control with unidirectional control input for uncertain nonlinear systems, Neural Computing and Applications, Vol. 30, No. 6, pp. 2003-2012, 2018.

[d] Structure control for the disturbance rejection in two electromechanical processes, Journal of the Franklin Institute, Vol. 353, No. 14, pp. 3610-3631, 2016.

2.- In the page 1, in the introduction, authors say "but this article doesdid not make a specific analysis of the controller performance.", it should be "but this article did not make a specific analysis of the controller performance."

3.- In the page 3, authors say " Based on the paper [20],it proposes an improved hierarchical sliding mode control.", in this part, they should include references [a], [b], [c], [d] which also consider the hierarchical sliding mode control.

4.- In the page 3, in the equation (3), authors use the sgn function which could produce the undesired chattering. They should clarify if they use a method the decrease the chattering.

6.- In the page 3, in the equation (3), authors should clarify that s + k should be different to zero in order to avoid singularities in the double power reaching law.

6.- In the page 4, in the equation (10), authors use the sgn function which could produce the undesired chattering. They should clarify if they use a method the decrease the chattering.

7.- In the page 4, in the equation (10), authors should clarify that s + k, p1, q2, p3, and q4 should be different to zero in order to avoid singularities in the controller.

8.- In the page 5, in the equation (16), authors should clarify that Jm and Jmd should be different to zero in order to avoid singularities in the controller.

9.- In the page 13, in the conclusions, authors should include some future research.

Reviewer 2 Report

the techniques the author presented in this article is not novel:

108: "a Gaussian function is introduced as smooth switching factor"  see for instance: DOI: 10.7763/IJIEE.2014.V4.482  and other references  the author claim that he used a novel hybrid control strategy to improve the performance of the controller, however the hybrid control strategy used is not novel. the author made some assumptions without elaborating to convince  the reader it is difficult sometimes to understand the objective of the author due to poor English structure.  

Reviewer 3 Report

Dear Authors; Editor;

This manuscript is presented a hybrid algorithm to control of flexible robot. However, the proposal is interesting but it has some issues which listed as follows:

How the authors can find the (S) dot? How proposed algorithm can improve the steady state performance and reduce the disturbance effects? Based on the proposed algorithm to improve the performance of IHSMC, the PID-IDA-PBC is introduced by authors to switching the control. The main question is why and how you use this algorithm? However, SMC is a robust and stable controller but the authors is used parallel technique namely, PID-IDA-PBC and the controller's output is applied to the IDA-PBC, so how the cascasde technique can improve the performance of SMC? (design the cascade controller is much difficult.) Why the title of this manuscript is hybrid?

Regards,

Round 2

Reviewer 3 Report

Dear Editor, Authors;

Regarding the authors' cover letter, the reviewer is suggested to more improve the revised manuscript. Thus, please more explanation about the following issues:

 Point 1: How the authors can find the (S) dot?

Point 3: Based on the proposed algorithm to improve the performance of IHSMC, the PID-IDA-PBC is introduced by authors to switching the control. The main question is why and how you use this algorithm?

Point 4: However, SMC is a robust and stable controller but the authors is used parallel technique namely, PID-IDA-PBC and the controller's output is applied to the IDA-PBC, so how the cascade technique can improve the performance of SMC?(design the cascade controller is much difficult.) [ how you can validate that this is robust?]

Point 5: Why the title of this manuscript is hybrid???

Round 3

Reviewer 3 Report

Dear Authors; Editor;

Thank you for your cover letter. Regarding the third round review, the manuscript can be suitable for publishing in this journal.

Good-Luck for your future research.

Regards